# DIVIDE AND NOT FORGET: ENSEMBLE OF SELECTIVELY TRAINED EXPERTS IN CONTINUAL LEARNING

**Grzegorz Rypeść**[1,2,*] **Sebastian Cygert**[1,3]**, Valeriya Khan**[1,2]**, Tomasz Trzciński**[1,2,4]**,**
**Bartosz Zieliński**[1,5] **& Bartłomiej Twardowski**[1,6,7]

[1]IDEAS-NCBR, [2]Warsaw University of Technology, [3] Gdańsk University of Technology,
[4] Tooploox, [5] Jagiellonian University, [6] Computer Vision Center, Barcelona
[7] Department of Computer Science, Universitat Autònoma de Barcelona,
`{grzegorz.rypesc, sebastian.cygert, valeriya.khan, tomasz.trzc`
`inski, bartosz.zielinski, bartlomiej.twardowski}@ideas-ncbr.pl`

## ABSTRACT

Class-incremental learning is becoming more popular as it helps models widen their applicability while not forgetting what they already know. A trend in this area is to use a mixture-of-expert technique, where different models work together to solve the task. However, the experts are usually trained all at once using whole task data, which makes them all prone to forgetting and increasing computational burden. To address this limitation, we introduce a novel approach named SEED. SEED selects only one, the most optimal expert for a considered task, and uses data from this task to fine-tune only this expert. For this purpose, each expert represents each class with a Gaussian distribution, and the optimal expert is selected based on the similarity of those distributions. Consequently, SEED increases diversity and heterogeneity within the experts while maintaining the high stability of this ensemble method. The extensive experiments demonstrate that SEED achieves state-of-the-art performance in exemplar-free settings across various scenarios, showing the potential of expert diversification through data in continual learning.

## 1 INTRODUCTION

In Continual Learning (CL), tasks are presented to the learner sequentially as a stream of non-i.i.d data. The model has only access to the data in the current task. Therefore, it is prone to *catastrophic forgetting* of previously acquired knowledge (French, 1999; McCloskey & Cohen, 1989). This effect has been extensively studied in Class Incremental Learning (CIL), where the goal is to train the classifier incrementally and achieve the best accuracy for all classes seen so far. One of the most straightforward solutions to alleviate forgetting is to store exemplars of each class. However, its application is limited, e.g., due to privacy concerns or in memory-constrained devices (Ravaglia et al., 2021). That is why more challenging, exemplar-free CIL solutions attract a lot of attention.

Many recent CIL methods that do not store exemplars rely on having a strong feature extractor right from the beginning of incremental learning steps. This extractor is trained on the larger first task, which provides a substantial amount of data (i.e., 50% of all available classes) (Hou et al., 2019; Zhu et al., 2022; Petit et al., 2023), or it starts from a large pre-trained model that remains unchanged (Hayes & Kanan, 2020a; Wang et al., 2022c) that eliminates the problem of representational drift (Yu et al., 2020). However, these methods perform poorly when little training data is available upfront. In Fig. 1, we illustrate both CIL setups, with and without the more significant first task. The trend is evident when we have a lot of data in the first task - results steadily improve over time. However, the

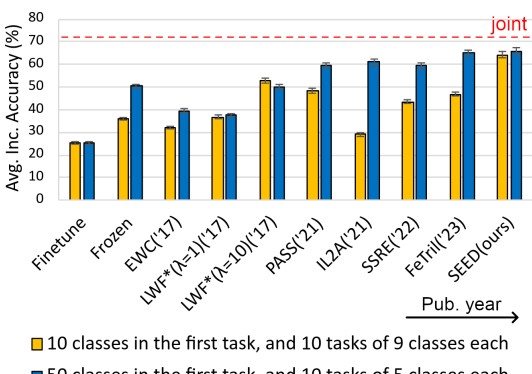

Figure 1: Exemplar-free Class Incremental Learning methods evaluated on CIFAR100 divided into eleven tasks for two different data distributions.

---

*Code: `https://github.com/grypesc/SEED`

progress is not evident for the setup with equal splits, where a frozen (or nearly frozen by high regularization) feature extractor does not yield good results. This setup is more challenging as it requires the whole network to continually learn new features (*plasticity*) and face the problem of catastrophic forgetting of already learned ones (*stability*).

One solution for this problem is architecture-based CIL methods, notably by expanding the network structure beyond a single model. Expert Gate (Aljundi et al., 2017) creates a new expert, defined as a neural network, for each task to mitigate forgetting. However, it can potentially result in unlimited growth in the number of parameters. Therefore, more advanced ensembling solutions, like CoSCL (Wang et al., 2022b), limit the computational budget using a fixed number of experts trained in parallel to generate features ensemble. In order to prevent forgetting, regularization is applied to all ensembles during training a new task, limiting their plasticity. Doan et al. (2022) propose ensembling multiple models for continual learning with exemplars for experience-replay. To perform efficient ensembling and control a number of the model's parameters, they enforce the model's connectivity to keep several ensembles fixed. However, exemplars are still necessary, and as in CoSCL, task-id is required during the inference.

As a remedy for the above issues, we introduce a novel ensembling method for exemplar-free CIL called *SEED: Selection of Experts for Ensemble Diversification*. Similarly to CoSCL and (Doan et al., 2022), SEED uses a fixed number of experts in the ensemble. However, only a single expert is updated while learning a new task. That, in turn, mitigates forgetting and encourages diversification between the experts. While only one expert is being trained, the others still participate in predictions. In SEED, the training does not require more computation than single-model solutions. The right expert for the update is selected based on the current ensemble state and new task data. The selection aims to limit representation drift for the classifier. The ensemble classifier uses multivariate Gaussian distribution representation associated with each expert (see Fig. 2). At the inference time, Bayes classification from all the experts is used for a final prediction. As a result, SEED achieves state-of-the-art accuracy for task-aware and task-agnostic scenarios while maintaining the high plasticity of the resulting model under different data distribution shifts within tasks.

In conclusion, the main contributions of our paper are as follows:

- We introduce SEED, a new method that leverages an ensemble of experts where a new task is selectively trained with only a single expert, which mitigates forgetting, encourages diversification between experts and causes no computational overhead during the training.
- We introduce a unique method for selecting an expert based on multivariate Gauss distributions of each class in the ensemble that limits representational drift for a selected expert. At the inference time, SEED uses the same rich class representation to perform Bayes classification and make predictions in a task-agnostic way.
- With the series of experiments, we show that existing methods that start CIL from a strong feature extractor later during the training mainly focus on stability. In contrast, SEED also holds high plasticity and outperforms other methods without any assumption of the class distribution during incremental learning sessions.

## 2   RELATED WORK

**Class-Incremental Learning (CIL)** represents the most challenging and prevalent scenario in the field of Continual Learning research (Van de Ven & Tolias, 2019; Masana et al., 2022), where during the evaluation task-id is unknown, and the classifier has to predict all classes seen so far. The simplest solution to fight catastrophic forgetting in CIL is to store exemplars, e.g. LUCIR (Hou et al., 2019), BiC (Wu et al., 2019), Foster (Wang et al., 2022a), WA (Zhao et al., 2020). Having exemplars greatly simplifies learning cross-task features. However, storing exemplars can not always be an option due to privacy issues or other limitations. Then, the hardest scenario *exemplar-free* CIL is considered, where number of methods exists: LwF (Li & Hoiem, 2016), SDC (Yu et al., 2020), ABD (Smith et al., 2021), PASS (Zhu et al., 2021b), IL2A (Zhu et al., 2021a), SSRE (Zhu et al., 2022), FeTrIL (Petit et al., 2023). Most of them favor stability and alleviate forgetting through various forms of regularization applied to an already well-performing feature extractor. Some approaches even concentrate solely on the incremental learning of the classifier while keeping the backbone network frozen (Petit et al., 2023). However, freezing the backbone can limit the plasticity and not be sufficient for more complex

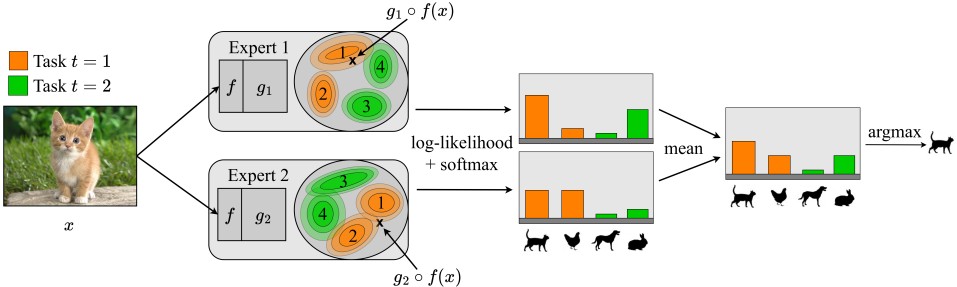

Figure 2: SEED comprises $K$ deep network experts $g_k \circ f$ (here $K = 2$), sharing the initial layers $f$ for higher computational performance. $f$ are frozen after the first task. Each expert contains one Gaussian distribution per class $c \in C$ in his unique latent space. In this example, we consider four classes, classes 1 and 2 from task 1 and classes 3 and 4 from task 2. During inference, we generate latent representations of input $x$ for each expert and calculate its log-likelihoods for distributions of all classes (for each expert separately). Then, we softmax those log-likelihoods and compute their average over all experts. The class with the highest average softmax is considered as the prediction.

settings, e.g., when tasks are unrelated, like in CTrL (Veniat et al., 2020). This work specifically aims at exemplar-free CIL, where the model's plasticity in learning new features for improved classification is still considered an essential factor.

**Growing architectures and ensemble.** Architecture-based methods for CIL can dynamically adjust some networks' parameters while learning new tasks, i.e. DER (Yan et al., 2021), Progress and Compress (Rusu et al., 2016) or use masking techniques, e.g. HAT (Serrà et al., 2018). In an extreme case, each task can have a dedicated expert network (Aljundi et al., 2017) or a single network per class (van de Ven et al., 2021). That greatly improves plasticity but also requires increasing resources as the number of parameters increases. Additionally, while the issue of forgetting is addressed, transferring knowledge between tasks becomes a new challenge. A recent method, CoSCL (Wang et al., 2022b), addresses this by performing an ensemble of a limited number of experts, which are diversified using a cooperation loss. However, this method is limited to task-aware settings. Doan et al. (2022) diversifies the ensemble by training tasks on different subspaces of models and then merging them. In contrast to our approach, the method requires exemplars to do so.

**Gaussian Models in CL.** Exemplar-free CIL methods based on cross-entropy classifiers suffer recency bias towards newly trained task (Wu et al., 2019; Masana et al., 2022). Therefore, some methods employ nearest mean classifiers with stored class centroids (Rebuffi et al., 2017; Yu et al., 2020). SLDA (Hayes & Kanan, 2020b) assigns labels to inputs based on the closest Gaussian, computed using the running class means and covariance matrix from the stream of tasks. In the context of continual unsupervised learning (Rao et al., 2019), Gaussian Mixture Models were used to describe new emerging classes during the CL session. Recently, in (Yang et al., 2021), a fixed, pre-trained feature extractor and Gaussian distributions with diagonal covariance matrices were used to solve the CIL problem. However, we argue that such an approach has low plasticity and limited applicability. Therefore, we propose an improved method based on multivariate Gaussian distributions and multiple experts that can learn new knowledge efficiently.

## 3 METHOD

The core idea of our approach is to directly diversify experts by training them on different tasks and combining their knowledge during the inference. Each expert contains two components: a feature extractor that generates a unique latent space and a set of Gaussian distributions (one per class). The overlap of class distributions varies across different experts due to disparities in expert embeddings. SEED takes advantage of this diversity, considering it both during training and inference.

**Architecture.** Our approach, presented in Fig. 2, consists of $K$ deep network experts $g_k \circ f$ for $k = 1, \dots, K$, sharing the initial layers $f$ for improving computational performance. $f$ are frozen after the first task. We consider the number of shared layers a hyperparameter (see Appendix A.3).

Moreover, each expert $k$ contains one Gaussian distribution $G_k^c = (\mu_k^c, \Sigma_k^c)$ per class $c$ for its unique latent space.

**Algorithm.** During inference, we perform an ensemble of Bayes classifiers. The procedure is presented in Fig. 2. Firstly, we generate representations of input $x$ for each expert $k$ as $r_k = g_k \circ f(x)$. Secondly, we calculate log-likelihoods of $r_k$ for all distributions $G_k^c$ associated with this expert

$$l_k^c(x) = -\frac{1}{2}[\ln(|\Sigma_k^c|) + S \ln(2\pi) + (r_k - \mu_k^c)^T (\Sigma_k^c)^{-1}(r_k - \mu_k^c)], \tag{1}$$

where $S$ is the latent space dimension. Then, we softmax those values $\widehat{l_k^1}, \ldots, \widehat{l_k^{|C|}} =$ softmax$(l_k^1, \ldots, l_k^{|C|}; \tau)$ per each expert, where $C$ is the set of classes and $\tau$ is a temperature. Class $c$ with the highest average value after softmax over all experts (highest $\mathbb{E}_k \widehat{l_k^c}$) is returned as a prediction for task agnostic setup. For task aware inference, we limit this procedure to classes from the considered task.

Our training assumes $T$ tasks, each corresponding to the non-overlapping set of classes $C_1 \cup C_2 \cup \cdots \cup C_T = C$ such that $C_t \cap C_s = \emptyset$ for $t \neq s$. Moreover, task $t$ is a training step with only access to data $D_t = \{(x, y) | y \in C_t\}$, and the objective is to train a model performing well both for classes of a new task and classes of previously learned tasks $(< t)$.

The main idea of training SEED, as presented in Fig. 3, is to choose and finetune one expert for each task, where the chosen expert should correspond to latent space where distributions of new classes overlap the least. Intuitively, this strategy causes latent space to change as little as possible, improving stability.

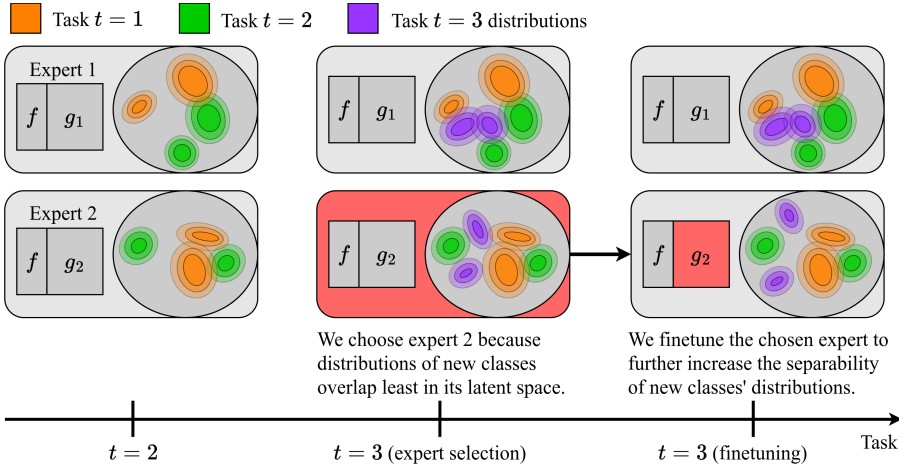

Figure 3: SEED training process for $K = 2$ experts, $T = 3$ tasks, and $|C_t| = 2$ classes per task. When the third task appears with novel classes $C_3$, we analyze distributions of $C_3$ classes (here represented as purple distributions) in latent spaces of all experts. We choose the expert where those distributions overlap least (here, expert 2). We finetune this expert to increase the separability of new classes further and move to the next task.

To formally describe our training, let us assume that we are in the moment of training when we have access to data $D_t = \{(x, y) | y \in C_t\}$ of task $t$ for which we want to finetune the model. There are two steps to take, selecting the optimal expert for task $t$ and finetuning this expert.

Expert selection starts with determining the distribution for each class $c \in C_t$ in each expert $k$. For this purpose, we pass all $x$ from $D_t$ with $y = c$ through deep network $g_k \circ f$. This results in a set of vectors in latent space for which we approximate a multivariate Gaussian distribution $q_{c,k}$. In consequence, each expert is associated with a set $Q_k = \{q_{1,k}, q_{2,k}, ..., q_{|C_t|,k}\}$ of $|C_t|$ distributions. We select expert $\bar{k}$ for which those distributions overlap least using symmetrized Kullback–Leibler divergence $d_{KL}$:

$$\bar{k} = \underset{k}{\operatorname{argmax}} \sum_{q_{i,k}, q_{j,k} \in Q_k} d_{KL}(q_{i,k}, q_{j,k}), \tag{2}$$

To finetune the selected expert $\bar{k}$, we add the linear head to its deep network and train $g_{\bar{k}}$ using $D_t$ set. As a loss function, we use cross-entropy combined with feature regularization based on knowledge distillation (Li & Hoiem, 2016) weighted with $\alpha$: $L = (1 - \alpha)L_{CE} + \alpha L_{KD}$, where $\mathcal{L}_{KD} = \frac{1}{|B|} \sum_{i \in B} ||g_{\bar{k}} \circ f(x_i) - g_{\bar{k}}^{old} \circ f(x_i)||$, $B$ is a batch and $g_{\bar{k}}^{old}$ is frozen $g_{\bar{k}}$.

While we use CE for its simplicity and effective clustering (Horiguchi et al., 2019), it can be replaced with other training objectives, such as self-supervision. Then, we remove the linear head, update distributions of $Q_{\bar{k}}$, and move to the next task.

Due to the random expert initializations, we skip the selection procedure for $K$ initial tasks and omit $L_{KD}$. Instead, we select the expert with the same number as the number task ($k = t$) and use $L = L_{CE}$. For the same reason, we calculate distributions of new tasks only for the experts trained so far ($k \leq t$). Finally, we fix $f$ after the first task so that finetuning one expert does not affect others.

## 4 EXPERIMENTS

In order to evaluate the performance of SEED and fairly compare it with other models, we utilize three commonly used benchmark datasets in the field of Continual Learning (CL): CIFAR-100 (Krizhevsky, 2009) (100 classes), ImageNet-Subset (Deng et al., 2009) (100 classes) and DomainNet (Peng et al., 2019) (345 classes, from 6 domains). DomainNet contains objects in very different domains, allowing us to measure models' adaptability to new data distributions. We create each task with a subset of classes from a single domain, so the domain changes between tasks (more extensive data drift). We always set $K = 5$ for SEED, so it consists of 5 experts. We evaluate all Continual Learning approaches in three different task distribution scenarios. We train all methods from scratch. Detailed information regarding experiments and the code are in the Appendix. We compare all methods with standard CIL evaluations using the classification accuracies after each task, and *average incremental accuracy*, which is the average of those accuracies (Rebuffi et al., 2017). We train all methods from scratch in all scenarios.

The first scenario is the CIL equal split setting, where each task has the same number of classes. This weakens the feature extractor trained on the first task, as there is little data. Therefore, this scenario better exposes the methods' plasticity. We reproduce results using FACIL(Masana et al., 2022), and PyCIL(Zhou et al., 2021) benchmarks for this setting. We train all methods using random crops, horizontal flips, cutouts, and AugMix (Hendrycks et al., 2019) data augmentations.

The second scenario is similar to the one used in (Hou et al., 2019), where the first task is larger than the subsequent tasks. This equips CIL methods with a more robust feature extractor than the equal split scenario. Precisely, the first task consists of either 50% or 40% of all classes. This setting allows methods that freeze the feature extractor (low plasticity) to achieve good results. We take baseline results for this setting from (Petit et al., 2023).

The third scenario is task incremental on equal split tasks (where the task id is known during inference). Here, the baseline results and numbers of models' parameters are taken from (Wang et al., 2022b). We perform the same data augmentations as in this work.

### 4.1 RESULTS

Tab. 1 presents the comparison of SEED and state-of-the-art exemplar-free CIL methods for CIFAR-100, DomainNet, and ImageNet-Subset in the equal split scenario. We report average incremental accuracies for various split conditions and domain shift scenarios (DomainNet). We present joint training as an upper bound for the CL training.

SEED outperforms other methods by a large margin in each setting. For CIFAR-100, SEED is better than the second-best method by 14.7, 17.5, and 15.6 percentage points for $T = 10, 20, 50$, respectively. The difference in results increases as there are more tasks in the setting. More precisely, for $T = 10$, SEED has 14.7 percentage points better accuracy than the second-best method (LwF*, which is LwF implementation with PyCIL (Zhou et al., 2021) data augmentations and learning rate schedule). At the same time, for $T = 50$ SEED is better by 15.6%. The results are consistent for other datasets, proving that SEED achieves state-of-the-art results in an equal split scenario. Moreover, based on DomainNet results, we conclude that SEED is also better in scenarios with a significant distributional shift. Detailed results for CIFAR100 T=50 and DomainNet T=36 are presented in

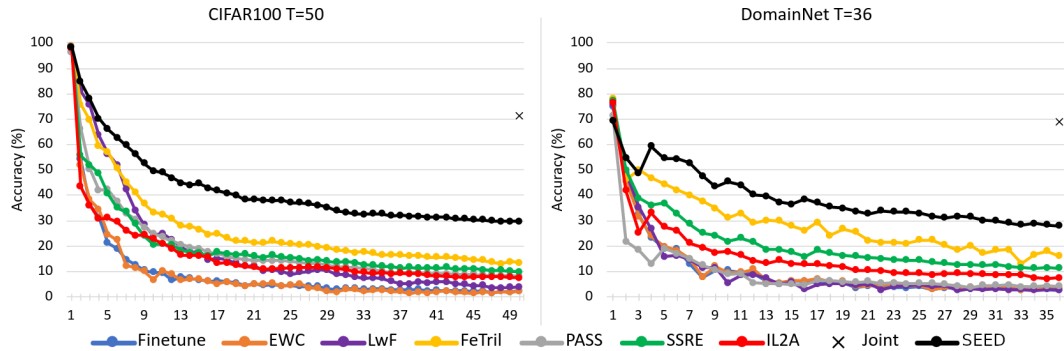

Figure 4: Class incremental accuracy achieved after each task for equal splits on CIFAR100 and DomainNet. SEED significantly outperforms other methods in equal split scenarios for many tasks (left) and more considerable data shifts (right).

Fig. 4. In this extreme setting, where each task consists of just little data, SEED results in significantly higher accuracies for the last tasks than other methods.

Table 1: Task-agnostic avg. inc. accuracy (%) for equally split tasks on CIFAR-100, DomainNet and ImageNet-Subset. The best results are in bold. SEED achieves superior results compared to other methods and outperforms the second best method (FeTrIL) by a large margin.

| CIL Method | CIFAR-100 (ResNet32) | | | DomainNet | | | ImageNet-Subset |
|---|---|---|---|---|---|---|---|
| | $T$=10 | $T$=20 | $T$=50 | $T$=12 | $T$=24 | $T$=36 | $T$=10 |
| Finetune | 26.4±0.1 | 17.1±0.1 | 9.4±0.1 | 17.9±0.3 | 14.8±0.1 | 10.9±0.2 | 27.4±0.4 |
| EWC (Kirkpatrick et al., 2017) (PNAS'17) | 37.8±0.8 | 21.0±0.1 | 9.2±0.5 | 19.2±0.2 | 15.7±0.1 | 11.1±0.3 | 29.8±0.3 |
| LwF* (Rebuffi et al., 2017) (CVPR'17) | 47.0±0.2 | 38.5±0.2 | 18.9±1.2 | 20.9±0.2 | 15.1±0.6 | 10.3±0.7 | 32.3±0.4 |
| PASS (Zhu et al., 2021b) (CVPR'21) | 37.8±1.1 | 24.5±1.0 | 19.3±1.7 | 25.9±0.5 | 23.1±0.5 | 9.8±0.3 | - |
| IL2A (Zhu et al., 2021a) (NeurIPS'21) | 43.5±0.3 | 28.3±1.7 | 16.4±0.9 | 20.7±0.5 | 18.2±0.4 | 16.2±0.4 | - |
| SSRE (Zhu et al., 2022) (CVPR'22) | 44.2±0.6 | 32.1±0.9 | 21.5±1.8 | 33.2±0.7 | 24.0±1.0 | 22.1±0.7 | 45.0±0.5 |
| FeTrIL (Petit et al., 2023) (WACV'23) | 46.3±0.3 | 38.7±0.3 | 27.0±1.2 | 33.5±0.6 | 33.9±0.5 | 27.5±0.7 | 58.7±0.2 |
| SEED | **61.7±0.4** | **56.2±0.3** | **42.6±1.4** | **45.0±0.2** | **44.9±0.2** | **39.2±0.3** | **67.8±0.3** |
| Joint | | 71.4±0.3 | | 63.7±0.5 | 69.3±0.4 | 69.1±0.1 | 81.5±0.5 |

**Large first task class incremental scenarios.** We present results for this setting in Tab. 2. For CIFAR-100, SEED is better than the best method (FeTrIL) by 4.6, 4.1, and 1.4 percentage points for $T = 6, 11, 21$, respectively. For $T = 6$ on ImageNet-Subset, SEED is better by 3.3 percentage points than the best method. However, with more tasks, $T = 11$ or $T = 21$, FeTrIL with a frozen feature extractor presents better average incremental accuracy.

We can notice that simple regularization-based methods such as EWC and LwF* are far behind more recent ones: FeTrIL, SSRE, and PASS, which achieve high levels of overall average incremental accuracy. However, these methods benefit from a larger initial task, where a robust feature extractor can be trained before incremental steps. In SEED, each expert can still specialize for a different set of tasks and continually learn more diversified features even with using regularization like LwF. The difference between SEED and other methods is noticeably smaller in this scenario than in the equal split scenario. This fact proves that SEED works better in scenarios where a strong feature extractor must be trained from scratch or where there is a domain shift between tasks.

**Task incremental with limited number of parameters.** We investigate the performance of SEED in task incremental scenarios. We compare it against another state-of-the-art task incremental ensemble method - CoSCL (Wang et al., 2022b) and follow the proposed limited number of models' parameters setup. We compare SEED to: HAT (Serrà et al., 2018), MARK (Hurtado et al., 2021), and BNS (Qin et al., 2021). Tab. 3 presents the results with the number of utilized parameters. Our method requires significantly fewer parameters than other methods and achieves better average incremental accuracy. Despite being designed to solve the exemplar-free CIL problem, SEED outperforms other task-incremental learning methods. Additionally, we check how the number of shared layers ($f$ function) affects SEED's performance. Increasing the number of shared layers decreases required parameters but negatively impacts task-aware accuracy. As such, the number of shared layers in SEED is a

Table 2: Comparison of CIL methods on ResNet18 and CIFAR-100 or ImageNet-Subset under larger first task conditions. We report task-agnostic avg. inc. accuracy from multiple runs. The best result is in bold. The discrepancy in results between SEED and other methods decreases compared to the equal split scenario.

| CIL Method | CIFAR-100 | | | ImageNet-Subset | | |
|---|---|---|---|---|---|---|
| | $T$=6 $\|C_1\|$=50 | $T$=11 $\|C_1\|$=50 | $T$=21 $\|C_1\|$=40 | $T$=6 $\|C_1\|$=50 | $T$=11 $\|C_1\|$=50 | $T$=21 $\|C_1\|$=40 |
| EWC* (Kirkpatrick et al., 2017) (PNAS'17) | 24.5 | 21.2 | 15.9 | 26.2 | 20.4 | 19.3 |
| LwF* (Rebuffi et al., 2017) (CVPR'17) | 45.9 | 27.4 | 20.1 | 46.0 | 31.2 | 42.9 |
| DeeSIL (Belouadah & Popescu, 2018) (ECCVW'18) | 60.0 | 50.6 | 38.1 | 67.9 | 60.1 | 50.5 |
| MUC* (Liu et al., 2020) (ECCV'20) | 49.4 | 30.2 | 21.3 | - | 35.1 | - |
| SDC* (Yu et al., 2020) (CVPR'20) | 56.8 | 57.0 | 58.9 | - | 61.2 | - |
| ABD* (Smith et al., 2021) (ICCV'21) | 63.8 | 62.5 | 57.4 | - | - | - |
| PASS* (Zhu et al., 2021b) (CVPR'21) | 63.5 | 61.8 | 58.1 | 64.4 | 61.8 | 51.3 |
| IL2A* (Zhu et al., 2021a) (NeurIPS'21) | 66.0 | 60.3 | 57.9 | - | - | - |
| SSRE* (Zhu et al., 2022) (CVPR'22) | 65.9 | 65.0 | 61.7 | - | 67.7 | - |
| FeTrIL* (Petit et al., 2023) (WACV'23) | 66.3 | 65.2 | 61.5 | 72.2 | **71.2** | **67.1** |
| SEED | **70.9±0.3** | **69.3±0.5** | **62.9±0.9** | **75.5±0.4** | 70.9±0.5 | 63.0±0.8 |
| Joint | | 80.4 | | | 81.5 | |

hyperparameter that allows for a trade-off between achieved results and the number of parameters required for training.

Table 3: Limited parameters setting on CIFAR-100 with random class order. The reported metric is average task aware accuracy (%). Results for SEED are presented for various numbers of shared layers. Although we designed SEED for the task agnostic setting, it achieves superior results to exemplar-free, architecture-based methods using fewer parameters.

| Approach | #Params. | 20-split | 50-split |
|---|---|---|---|
| HAT | 6.8M | 77.0 | 80.5 |
| MARK | 4.7M | 78.3 | - |
| BNS | 6.7M | - | 82.4 |
| CoSCL(EWC+LWF) | 4.6M | 79.4±1.0 | 87.9±1.1 |
| SEED | 3.2M | **86.8±0.3** | **91.2±0.4** |
| SEED(1 shared) | 3.2M | 86.7±0.6 | 91.2±0.5 |
| SEED(11 shared) | 3.1M | 85.6±0.3 | 89.6±0.2 |
| SEED(21 shared) | **2.7M** | 82.4±0.4 | 88.1±0.5 |

Table 4: Ablation study of SEED for CIL setting with T=10 on ResNet32 and CIFAR-100. Avg. inc. acc. is reported. Multiple components of SEED were ablated. SEED as-designed presents the best performance.

| Approach | Acc.(%) |
|---|---|
| SEED(5 experts) | **61.7 ±0.4** |
| standard ensemble | 56.9±0.4 |
| weighted ensemble | 57.0±0.5 |
| CoSCL ensemble | 57.3±0.4 |
| w/o multivariate Gauss. | 53.5±0.5 |
| w/o covariance | 54.1±0.3 |
| w/o temp. in softmax | 59.2±0.5 |
| w/ ReLU | 57.8±0.6 |

## 4.2 DISCUSSION

**Is SEED better than other ensemble methods?** We want to verify that the improved performance of our method comes from more than just forming an ensemble of classifiers. Hence, we compare SEED with the vanilla ensemble approach to continual learning, where all experts are initialized with random weights, trained on the first task, and sequentially fine-tuned on incremental tasks. The final decision is obtained by averaging the predictions of ensemble members. We present results in Tab. 4. Using the standard ensemble decreases the accuracy by 4.8%. We also experiment with the approaches where the predictions are weighted during inference by the confidence of the ensemble members (using prediction entropy, as in (Ruan et al., 2023) and where the experts are trained with additional *ensemble cooperation* loss from (Wang et al., 2022b). However, they yielded similar results to uniform weighting.

**Diversity of experts** Fig. 5 and Fig.11 (Appendix) depict the quality of each expert on various tasks and their respective contributions to the ensemble. It can be observed that experts specialize in tasks on which they were fine-tuned. For each task, there is always an expert who exhibits over 2.5% points better accuracy than the average of all experts. This demonstrates that experts specialize in different tasks. Additionally, the ensemble consistently achieves higher accuracy (ranging from 6% to 10% points) than the average of all experts on all tasks. Furthermore, the ensemble consistently outperforms the best individual expert, indicating that each expert contributes uniquely to the ensemble. See the

details in Fig. 10 (Appendix) for more analysis of overlap strategy from Eq. 2 that also presents how experts are diversified between the tasks.

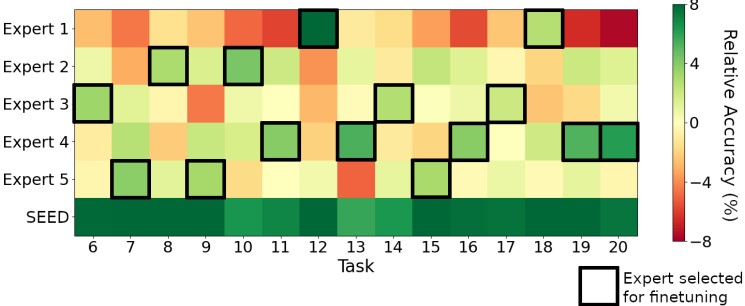

Figure 5: Diversity of experts on CIFAR-100 dataset with $T = 20$ split. The presented metric is relative accuracy (%) calculated by subtracting the accuracy of each expert from the averaged accuracy of all experts. Black squares represent experts selected to be finetuned on a given task. Although we do not impose any cost function associated with experts' diversity, they tend to specialize in different tasks by the design of our method. Moreover, our ensemble (bottom row) always performs better than the best expert, proving that each expert contributes uniquely to the ensemble in SEED.

**Expert selection strategy.** In order to demonstrate that our minimum overlap selection strategy (KL-max) improves the performance of the SEED architecture, we compare it to three other selection strategies. The first is a random selection strategy, where each expert has an equal probability of being chosen for finetuning. The second is a round-robin selection strategy, where for a task $t$, an expert with no. $1 + (t - 1 \bmod K)$ is chosen for a finetuning. The third one is the maximum overlap strategy (KL-min), in which we choose the expert for which the overlap between latent distributions of new classes is the highest. We conduct ten runs on CIFAR-100 with a Resnet32 architecture, three experts, and a random class order and report the average incremental accuracy in Fig. 6. Our minimum overlap selection strategy shows a higher mean and median than the other methods.

**Ablation study.** In Tab. 4, we present the ablation study for SEED. We report task-agnostic inc. avg. accuracy for five experts on CIFAR-100 and ResNet32, where results are averaged over three runs. Firstly, we remove or replace particular SEED components. We start by replacing the multivariate Gaussian distribution with its diagonal form. This reduces accuracy to 53.5%. Then, we remove Gaussian distributions and represent each class as a mean prototype in the latent space and use Nearest Mean Classifier (NMC) to make predictions. This also reduces accuracy, which shows that using multivariate distribution is important for SEED accuracy. Secondly, we check the importance of using temperature in the softmax function during inference. SEED without temperature ($\tau = 1$) achieves worse results than with temperature ($\tau = 3$), allowing more experts to contribute to the ensemble with more fuzzy likelihoods. At last, we analyze various SEED modifications, i.e., adding ReLU activations (like in the original ResNet) at the last layer, which decreased the accuracy by 3.9% points. It is because it is easier for the neural network trained with cross-entropy loss to represent features as Gaussian distribution if nonlinear activation is removed. See Tab. 7 for additional ablation study.

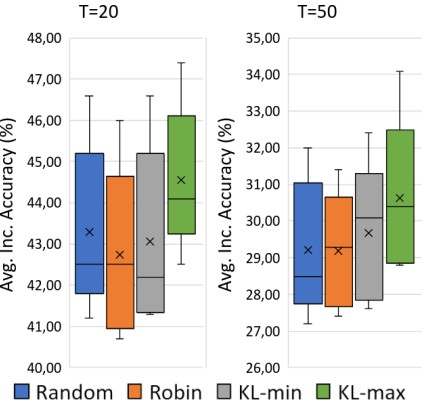

Figure 6: Avg. inc. accuracy of 10 runs with different class orderings for CIFAR-100 and different fine-tuning expert selection strategies for $T = 20, 50$ and three experts. Our KL-max expert selection strategy yields better results than random, round-robin, and KL-min.

**Plasticity vs stability trade-off.** SEED uses feature distillation in trained expert to alleviate forgetting. To assess the influence of the regularization on overall method performance, we use the forgetting and intransigence measures defined in (Chaudhry et al., 2018). Fig. 7 (left) shows the relationship between forgetting and intransigence for four different regularization-based methods: SEED, EWC as a parameters regularization-based method, LWF as regularization of network's output with distillation,

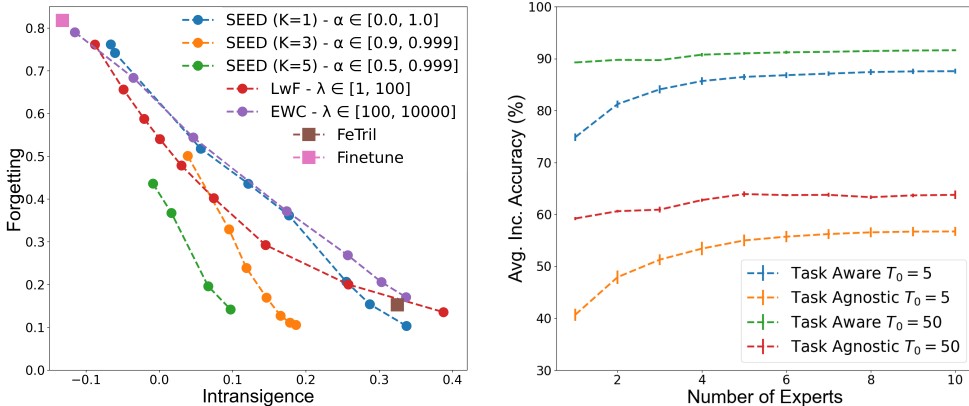

Figure 7: CIFAR-100. (Left) Forgetting and intransigence for different methods when manipulating the stability-plasticity parameters for $T = 10$. SEED with 5 experts achieves the best forgetting-intransigence trade-off. (Right) SEED accuracy as a function of a number of experts for $T = 20$ with 5 or 50 classes in the first task. Bars reflect standard dev. out of three runs.

and a recent FeTrIL method (Petit et al., 2023). For SEED, we adapt plasticity using the $\alpha$ and $K$ parameter, and for both EWC and LWF, we change the $\lambda$ parameter. FeTrIL method has no such parameter, as it uses a frozen backbone. The trade-off between stability and plasticity is evident. The FeTrIL model is very intransigent, with low plasticity and low forgetting. Plasticity is crucial in the CIL setting with ten or more tasks with an equal number of classes. Thus, EWC and LwF, need to be less rigid and exhibit more forgetting. The SEED model for $K = 3$ and $K = 5$, achieves much better results than FeTrIL while remaining less intransigent and more stable than LwF and EWC. By adjusting the $\alpha$ trade-off parameter of SEED, its stability can be controlled for any number of experts.

**Number of experts.** In Fig. 7 (right), we analyze how the number of experts influences the avg. incremental accuracy achieved by SEED. Changing the number of experts from 1 to 5 increases task-agnostic and task-aware accuracy by $\approx 15\%$ for $T_0 = 5$. However, for $T_0 = 50$, the increase is less significant ( 2% and  5% for task aware and task agnostic settings, respectively). These results suggest that scenarios with the significantly bigger first task are simpler than equal split ones. Moreover, going beyond five experts does not improve final CIL performance so much.

## 5 CONCLUSIONS

In this paper, we introduce an exemplar-free CIL method called SEED. It consists of a limited number of trained from scratch experts that all cooperate during inference, but in each task, only one is selected for finetuning. Firstly, this decreases forgetting, as only a single expert model's parameters are updated without changing learned representations of the others. Secondly, it encourages diversified class representations between the experts. The selection is based on the overlap of distributions of classes encountered in a task. That allows us to find a trade-off between model plasticity and stability. Our experimental study shows that the SEED method achieves state-of-the-art performance across several exemplar-free class-incremental learning scenarios, including different task splits, significant shifts in data distribution between tasks, and task-incremental settings. In the ablation study, we proved that each SEED component is necessary to obtain the best results.

**Reproducibility and limitations of SEED** We enclose the code in the supplementary material, and results can be reproduced by following the readme file. Our method has three limitations. Firstly, SEED may be not feasible for scenarios where tasks are completely unrelated and the number of parameters is limited, as in that case sharing initial parameters between experts may lead to a poor performance. Secondly, SEED requires the maximum number of experts given upfront, which can be found as a limitation of our method for new settings. Thirdly, calculating a distribution for a class may not be possible if the class's covariance matrix is singular. We address the last problem by decreasing latent space size. We elaborate more on this in the Appendix A.2.

ACKNOWLEDGMENTS

This research was supported by National Science Centre, Poland grant no 2020/39/B/ST6/01511, grant no 2022/45/B/ST6/02817, grant no 2022/47/B/ST6/03397, PL-Grid Infrastructure grant nr PLG/2022/016058, and the Excellence Initiative at the Jagiellonian University. This work was partially funded by the European Union under the Horizon Europe grant OMINO (grant number 101086321) and Horizon Europe Program (HORIZON-CL4-2022-HUMAN-02) under the project "ELIAS: European Lighthouse of AI for Sustainability", GA no. 101120237, it was also co-financed with funds from the Polish Ministry of Education and Science under the program entitled International Co-Financed Projects. Bartłomiej Twardowski acknowledges the grant RYC2021-032765-I. Views and opinions expressed are however those of the author(s) only and do not necessarily reflect those of the European Union or the European Research Executive Agency. Neither the European Union nor European Research Executive Agency can be held responsible for them.

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

# A APPENDICES

## A.1 IMPLEMENTATION DETAILS

All experiments are the average over three runs and all methods are trained from scratch as in their original papers. We implemented SEED in FACIL (Masana et al., 2022) framework using Python 3 programming language and PyTorch (Paszke et al., 2019) machine learning library. We utilized a computer equipped with AMD EPYC$^{TM}$ 7742 CPU and NVIDIA A-100$^{TM}$ GPU to perform experiments. On this machine, SEED takes around 1 hour to be trained on CIFAR100 for $T = 10$.

For all experiments, SEED is trained using the Stochastic Gradient Descent (SGD) optimizer for 200 epochs per task, with a momentum of $0.9$, weight decay factor equal $0.0005$, $\alpha$ set to $0.99$, $\tau$ set to $3$ and an initial learning rate of $0.05$. The learning rate decreases ten times after 60, 120, and 160 epochs. As the knowledge distillation loss, we employ the L2 distance calculated for embeddings in the latent space. We set the default number of experts to 5 and class representation dimensionality $S$ to 64. In order to find the best hyperparameters for SEED, we perform a manual hyperparameter search on a validation dataset.

**Tab. 1, Fig. 4 and Fig. 9**. We perform experiments for all methods using implementations provided in FACIL and PyCIL (Zhou et al., 2021) frameworks. We use ResNet32 as a feature extractor for CIFAR100 and ResNet18 for DomainNet and ImageNet-Subset. For DomainNet $T = 12$, we use 25 classes per task; for $T = 18$, we use 10; for $T = 36$, we use 5.

All methods were trained using the same data augmentations: random crops, horizontal flips, cutouts, and AugMix (Hendrycks et al., 2019). For baseline methods, we set default hyperparameters provided in benchmarks. However, for LwF, we use $\lambda = 10$ as we observed that this significantly improved its performance.

**Tab. 2**. For baseline results, we provide results reported in (Petit et al., 2023). All CIL methods use the same data augmentations: random resized crops, horizontal flips, cutouts, and AugMix (Hendrycks et al., 2019).

**Tab. 3**. The setting and baseline results are identical to (Wang et al., 2022b). We train SEED with the same data augmentation methods as other methods: horizontal flips and random crops. Here, we use five experts consisting of the Resnet32 network.

**Fig. 6** We calculate relative accuracy by subtracting each expert's accuracy from the average accuracy of all experts as in (Wang et al., 2022b). We perform 10 runs with random seeds.

**Fig. 7**. Below we report the range of used parameters for plotting the forgetting-intransigence curves (Fig. 7 - left).

- LwF: $\lambda \in \{1, 2, 3, 5, 7, 10, 15, 20, 25, 50, 100\}$
- EWC: $\lambda \in \{100, 500, 1000, 2500, 5000, 7500, 10000\}$
- SEED $K = 1$: $\gamma \in \{0.0, 0.25, 0.5, 0.9, 0.95, 0.97, 0.99, 0.999\}$
- SEED $K = 3$: $\gamma \in \{0.9, 0.95, 0.96, 0.97, 0.98, 0.99, 0.999\}$
- SEED $K = 5$: $\gamma \in \{0.5, 0.9, 0.97, 0.999\}$

## A.2 RESNET ARCHITECTURE MODIFICATION

The backbone for the SEED method can be any popular neural network with its head removed. This study focuses on a family of modified ResNet (He et al., 2016) architectures.

ResNet architecture is a typical neural architecture used for the continual learning setting. In this work, we follow this standard. However, there are two minor changes to ResNet in our algorithm.

Due to ReLU activation functions placed at the end of each ResNet block, latent feature vectors of ResNet models consist of non-negative elements. That implies that every continuous random variable representing a class is defined on $[0; \infty)^S$, where $S$ is the size of a latent vector. However, Gaussian distributions are defined for random variables of real values, which, in our case, reduces the ability to represent classes as multivariate Gaussian distributions. In order to alleviate this problem, we remove the last ReLU activation function from every block in the last layer of ResNet architecture.

Secondly, the size $S$ of the latent sample representation should be adjustable. There are two reasons for that. Firstly, if $S$ is too big and a number of class samples is low, $\Sigma$ can be a singular matrix. This implies that the likelihood function might not be well-defined. Secondly, adjusting $S$ allows us to reduce the number of parameters the algorithm requires. We overcome this issue by adding a 1x1 convolution layer with $S$ kernels after the last block of the architecture. For example, this allows us to represent feature vectors of Resnet18 with 64 elements instead of 512.

### A.3 MEMORY FOOTPRINT

SEED requires:

$$|\theta_f| + K|\theta_g| + \sum_{i=1}^{K} \sum_{j=i}^{T} |C_j|(S + \frac{S(S+1)}{2}) \tag{3}$$

parameters to run, where $|\theta_f|$ and $|\theta_g|$ represent number of parameters of $f$ and $g$ functions, respectively. $S$ is dimensionality of embedding space, $K$ is number of experts, $T$ is number of tasks, $|C_j|$ is a number of classes in $j$-th task.

This total number of parameters used by SEED can be limited in several ways:

- Decreasing S by adding a convolutional 1x1 bottleneck layer at the network's end.
- Pruning parameters.
- Performing weight quantization.
- Using simpler feature extractor.
- Increasing number of shared layers (moving parameters from $g$ into $f$ function).
- Simplifying multivariate Gaussian distributions to diagonal covariance matrix or prototypes.

### A.4 NUMBER OF PARAMETERS VS ACCURACY TRADE-OFF

To investigate the trade-off between the number of the SEED's parameters and the overall average incremental accuracy of the method, we conducted several experiments with a different number of experts and shared layers (as in a previous experiment in Tab. 3 we only adjust the number of layers). We see that these two factors indeed control and decrease the number of required parameters, e.g., sharing the first 25 layers in Resnet32 decreases memory footprint by 0.8 million parameters when we use five experts. However, it slightly hurts the performance of SEED, as the overall average incremental accuracy drops by 4.4%. These results, combined with expected forgetting/intransigence, can guide an application of SEED for a particular problem.

We additionally compare SEED to the best competitor - FeTrIL (with Resnet32) in a low parameters regime. FeTrIL stores feature extractor without the linear head and prototypes of each class. For SEED we utilize Resnet20 network with a number of kernels changed from 64 to 48 in the third block. We use 5 experts which share first 17 layers. We present results in Tab. 5. We present the number of network weights and total number of parameters which for SEED also includes multivariate Gaussian distributions. SEED has 13K less parameters than FeTrIL but achieves better accuracy on 3 settings.

Table 5: SEED outperforms competitors in terms of performance and number of parameters on equally split CIFAR100, however it requires decreasing size of the feature extractor network and sharing first 17 layers.

| CIL Method | Network | Network weights | Total params. | CIFAR-100 | | |
|---|---|---|---|---|---|---|
| | | | | $T=10$ | $T=20$ | $T=50$ |
| EWC* (Kirkpatrick et al., 2017) (PNAS'17) | ResNet32 | 473K | 473K | 24.5 | 21.2 | 15.9 |
| LwF* (Rebuffi et al., 2017) (CVPR'17) | Resnet32 | 473K | 473K | 45.9 | 27.4 | 20.1 |
| FeTrIL (Petit et al., 2023) (WACV'23) | Resnet32 | 473K | 473K | 46.3±0.3 | 38.7±0.3 | 27.0±1.2 |
| SEED | Resnet20* | 339K | 460K | 54.7±0.2 | 48.6±0.3 | 33.1±1.1 |

### A.5 PRETRAINING SEED

We study the impact of using a pretrained network with SEED on its performance. For this purpose we utilize ResNet-18 with weights pretrained on ImageNet-1K as a feature extractor for every expert.

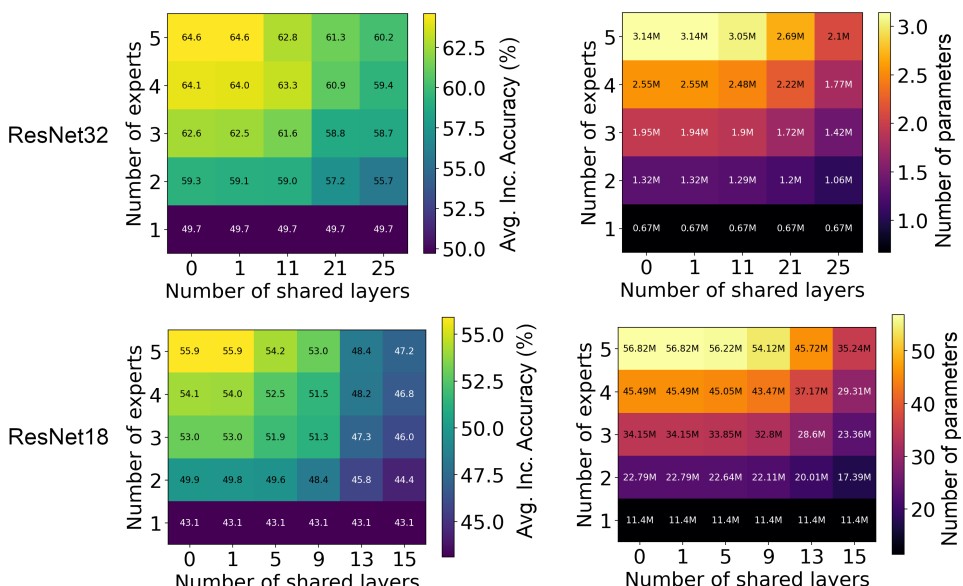

Figure 8: Impact of number of experts and number of shared layers on accuracy and number parameters of SEED. We utilize CIFAR100 with $|T| = 10$. We can observe that accuracy drops when decreasing the number of experts and increasing the number of shared layers.

| DomainNet | Avg. Inc. Accuracy (%) | | Forgetting (%) | |
|:---:|:---:|:---:|:---:|:---:|
| $|T|$ | From scratch | Pretrained | From scratch | Pretrained |
| 12 | 45.0 | 53.1 | 12.1 | 12.8 |
| 24 | 44.9 | 54.2 | 11.2 | 12.1 |
| 36 | 39.2 | 53.6 | 13.7 | 15.6 |

Table 6: Pretraining experts in SEED increases its accuracy while slightly increasing forgetting. We compare ResNet-18 with weights pretrained on ImageNet-1K to a randomly initilized ResNet-18 as feature extractors of each expert.

We compare it to SEED initialized with random weights (training from scratch) in Tab.6. Pretrained SEED achieves better average incremental accuracy by: $8.1\%, 9.3\%, 14.4\%$ on DomainNet split to 12, 24, 36 tasks respectively.

A.6    ADDITIONAL RESULTS

In this section we provide more experimental results. Fig. 10 presents insight into diversity of experts on CIFAR100 for $T = 50$ and 5 experts. We measure value of overlap per expert in each task given by Eq. 2. Average value of the function differ between tasks, e.g., for task 49. it equals to $\approx 3.5$, while for task 33. it equals to $\approx 18.0$. This is due to semantic similarity between classes in a given task, classes in task 49. (cups, bowls) are semantically more similar then classes in task 33. (bed, dolphin). However, in each task we can observe significant variance between values for experts. This proves that classes overlap differently in experts, therefore experts are diversified what allows SEED to achieve great results.

In Fig. 9, we present accuracies obtained after equal split tasks for Table 1. We report results for CIFAR100 and DomainNet datasets. We can observe that for DomainNet SEED achieves $15\%$ better accuracy after the last incremental step than FeTriL. On CIFAR100 SEED achieves $20\%$ and $16\%$ better accuracy than the best method. This proves that SEED achieves superior results to state-of-the-art methods on equal-split and big domain shift settings.

Table 7 presents an additional ablation study for SEED. We test various ways to approximate class distributions in the latent space on CIFAR100 dataset and $T = 10$. Firstly, we replace multivariate Gaussian distribution with a Gaussian Mixture Model (GMM) consisting of 2 multivariate Gaussian

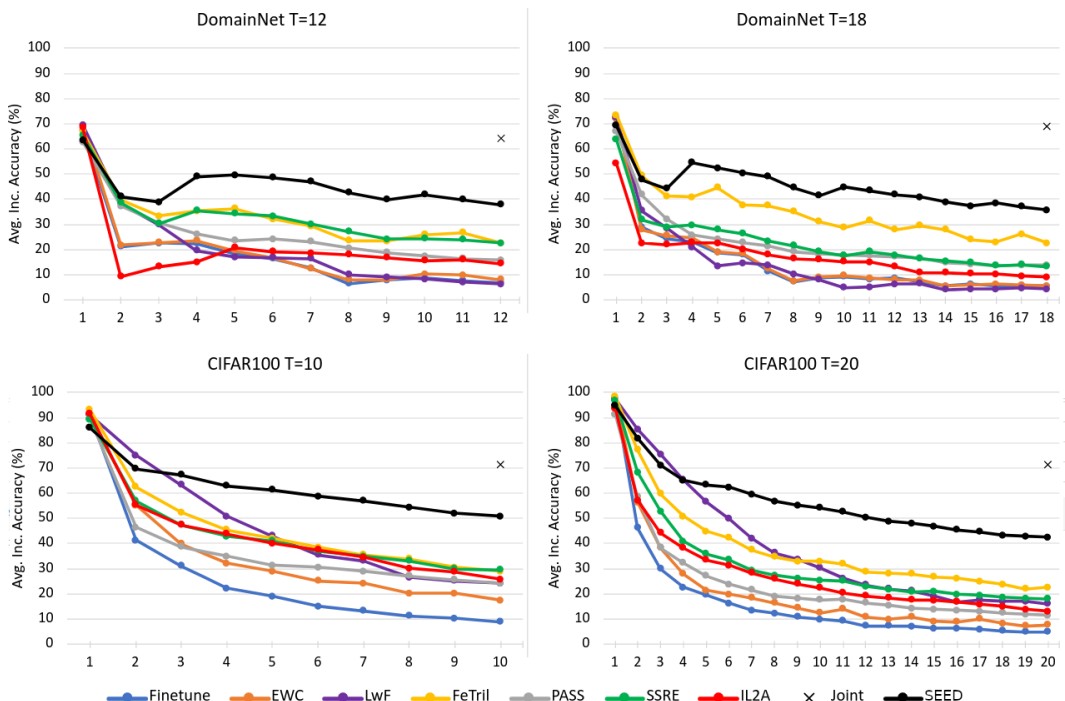

Figure 9: Accuracy after each task for equal splits on CIFAR100 and DomainNet. SEED significantly outperforms other methods in equal split scenarios for many tasks (top) and more considerable data shifts (bottom).

distributions. It slightly reduces the accuracy (by 1.2%). Then, we abandon multivariate Gaussians and approximate classes using 2 and 3 Gaussian distributions with the diagonal covariance matrix. That decreases accuracy by a large margin. We also approximate classes using their prototypes (centroids) in the latent space. This also reduces the performance of SEED.

Table 7: Ablation study of SEED for CIL setting with T=10 on ResNet32 and CIFAR-100. Avg. inc. accuracy is reported. We test different variations of class representation (such as Gaussian Mixture Model, diagonal covariance matrix or prototypes). SEED presents the best performance when used as designed.

| Approach | Acc.(%) |
|---|---|
| SEED(5 experts) | **61.7** ±0.5 |
| w/ 2× multivariate | 60.5±0.7 |
| w/ 2× diagonal | 53.8 ±0.1 |
| w/ 3× diagonal | 53.8±0.3 |
| w/ prototypes | 54.1±0.3 |

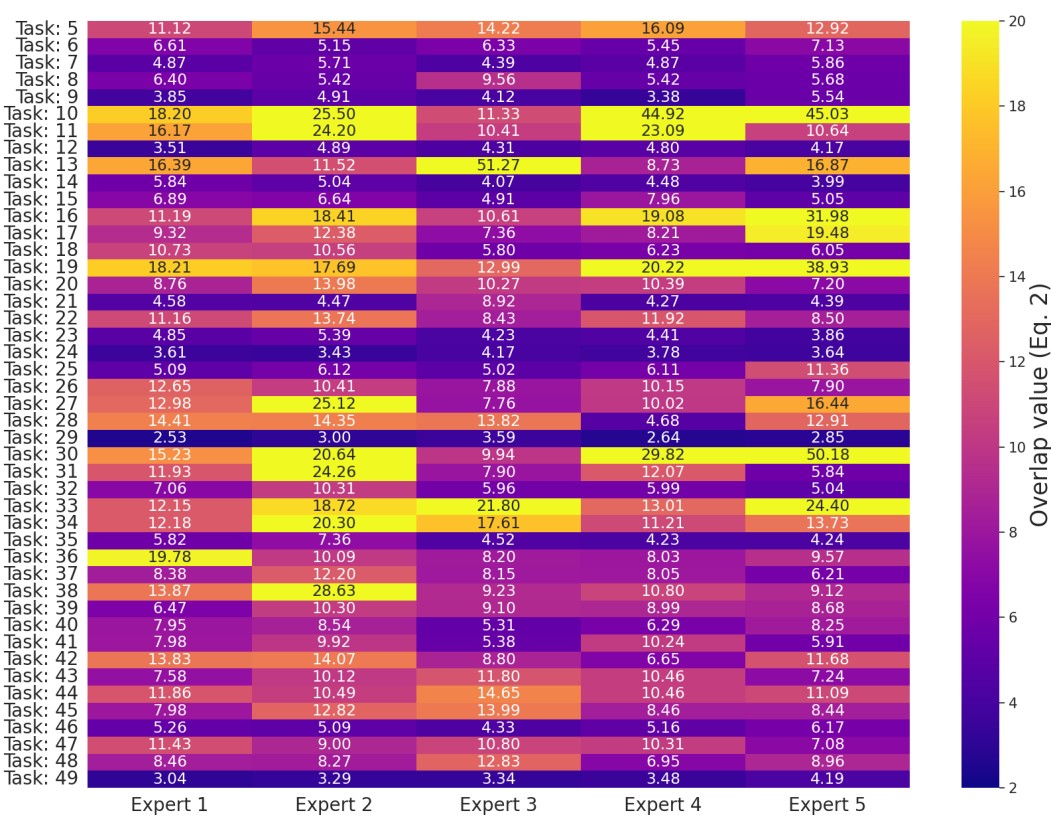

Figure 10: Overlap of class distributions in each task per expert on CIFAR-100 dataset with $T = 50$ split and random class order. Diversification by data yields high variation in overlap values in each task what proves that experts are diversified and learn different features. Average overlap values differ between tasks, as they depend on semantic similarity of classes. Classes in task 49. are semantically similar (cups, bowls) but classes in task 33. are different (beds, dolphins).

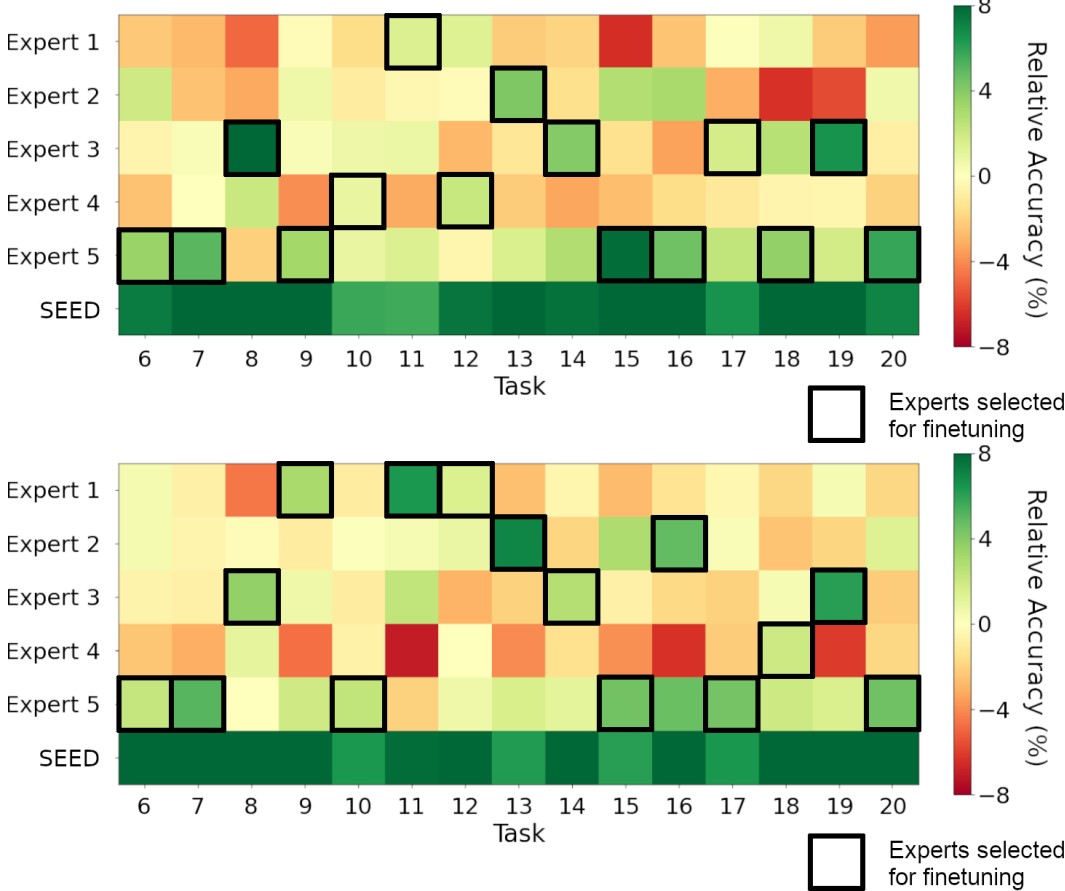

Figure 11: Diversity of experts on CIFAR-100 dataset with $T = 20$ split, different seeds and random class order. The presented metric is relative accuracy (%). Black squares represent experts selected to be finetuned on a given task. We can observe that experts specialize in different tasks.

