# OpenReview forum: "Divide and not forget: Ensemble of  selectively trained experts  in Continual Learning"
_ICLR.cc/2024/Conference — ICLR 2024 poster_

### Official Review · Reviewer_rq4G · 2023-10-29

**Soundness:** 3 good
**Presentation:** 3 good
**Contribution:** 3 good
**Rating:** 6
**Confidence:** 3

**Summary:**

This paper presents Selection of Experts for Ensemble Diversification (SEED), an algorithm that tackles exemplar-free class-incremental learning (CIL). Existing exemplar-free CIL algorithms incrementally trains new local experts on new tasks, and SEED provides a task selection technique to diversify trained experts, mitigate forgetting, and maintain plasticity. Specifically, an expert is a Bayes classifier that targets a distinct set of classes, with each class corresponding to a Gaussian distribution. To diversify expert selection, the next task is selected via the highest KL-divergence from encountered distributions. Inference is done by averaging the logits output from each expert and selecting the class with highest probability.

**Strengths:**

1. The paper clearly identifies a problem of task diversification in CIL.
2. The language is easy to follow.

**Weaknesses:**

1. Lack of novelty: selecting the next task based on largest KL divergence in continual learning is not a novel strategy. So is the expandable MoE architecture and the inference method.
2. Overhead growth: one biggest disadvantage of architecture expansion is that the memory overhead grows linearly with respect to the number of tasks. That is, it needs K experts for K tasks with nothing reusable. SEED omits the selection procedure to decide the number of experts needed even if the number of tasks is large. This is not scalable in continual learning, where the number of tasks can be unbounded.

**Questions:**

Please address the two weakness points above.

---

> ### Comment · Reviewer_3XPS · 2023-11-13
>
> Can you be a bit more elaborate on Weakness 1 to give the authors an opportunity to answer and the other reviewers to verify for themselves? In particular, I'd like to see references to the existing work. Thanks!

---

> ### Author Response · Authors · 2023-11-15
>
> We appreciate the feedback provided by the Reviewer. We will now address the specific weaknesses indicated:
>
>
> **Novelty concern related to the next task selection**
>
> > *Selecting the next task based on largest KL divergence in continual learning is not a novel strategy. So is the expandable MoE architecture and the inference method.*
>
> We thank the reviewer for that comment, yet we believe that there may be some misunderstanding of our method. Please note that we do not select tasks. We use KL-divergency to assign a single, existing expert to be trained on a given task. However, during the inference time, we use an ensemble of all available $K$ experts whose predictions are aggregated, and this approach is different from conventional MoE architectures. As far as the novelty of our approach is concerned, we believe that our work is the first to show how allocating experts to be trained on tasks can alleviate catastrophic forgetting in continual learning. We would appreciate if the Reviewer could link references to the mentioned relevant existing continual learning strategies so we can refer to them in our work.
>
> **Concern of the overhead growth**
>
> As stated in Method section 3 ```Our approach, presented in Fig.2, consists of $K$ deep network experts```. Therefore, K is the number of experts, which is equal to 3 or 5 in our experimental section regardless of the number of tasks |T|. We do not grow the network architecture above that number, e.g., in Fig. 4 and Tab. 1 we present experiments with |T|=50, where K=5 experts are used.
>
> ---
>
> We hope our explanation alleviates any concerns the Reviewer may have. However, if there are any remaining uncertainties, kindly specify references and additional questions, and we can further discuss them. Otherwise, we'd appreciate if the reviewer reconsidered the final score of our submission.

---

> ### Author Response · Authors · 2023-11-21
>
> Dear Reviewer rq4G,
>
> The discussion period ends in 24 hours.
>
> We appreciate the Reviewer's contributions to the reviewing process. We have put much effort into our work and would appreciate the Reviewer's constructive feedback.
>
> We are also dedicated to the review process and ensuring that our work adheres to the highest standards. We kindly ask the Reviewer to respond to our previous comment and identify specific areas in our submission where further improvement would allow the Reviewer to confidently recommend our paper for acceptance (more than above-borderline acceptance). We highly value and appreciate your efforts.

---

> > ### Comment · Reviewer_rq4G · 2023-11-21
> >
> > Thank you for your response. By the novelty concern, I mean that there exist selection algorithms in continual learning by measuring task similarities, e.g. [1, 2], and one of the used method of measuring similarities is by KL-divergence [3]. Moreover, using experts to model distributions at each task in continual learning is also an established work [4].
> >
> > If the authors are able to elaborate the main differences from a combination of experts and task similarity measurement, and convince me that the difference is non-trivial, I would be happy to raise my rating to 6 or above.
> >
> > [1] Basterrech, Sebastián, and Michał Woźniak. "Tracking changes using Kullback-Leibler divergence for the continual learning." 2022 IEEE International Conference on Systems, Man, and Cybernetics (SMC). IEEE, 2022.
> >
> > [2] Simon, Christian, Piotr Koniusz, and Mehrtash Harandi. "On learning the geodesic path for incremental learning." Proceedings of the IEEE/CVF conference on Computer Vision and Pattern Recognition. 2021.
> >
> > [3] Pentina, Anastasia, and Christoph H. Lampert. "Lifelong learning with non-iid tasks." Advances in Neural Information Processing Systems 28 (2015).
> >
> > [4] Lee, Soochan, et al. "A neural dirichlet process mixture model for task-free continual learning." arXiv preprint arXiv:2001.00689 (2020).

---

> > > ### Author Response · Authors · 2023-11-21
> > >
> > > We express our gratitude to the Reviewer for their insightful response and active participation in the discussion, including the incorporation of interesting references to other works. We have thoroughly reviewed and considered them while preparing this answer.
> > >
> > > First, we sum up the referenced works and compare them to ours:
> > >
> > > Work [1] is an analysis of monitoring changes in the probabilistic distribution of multi-dimensional data streams and predicts concept drift based on Kullback-Leibler divergence (KLD). But, the work itself does not tackle the problem of exemplar-free class incremental learning. Its focus is on tracking changes in generic data streams. The experimental section presents the results of using KLD only for four or six feature data streams with binary outputs. This work indeed confirms that KL-divergency can be a good choice for checking distribution drift and can support the use of KLD in our method.
> > >
> > > In [2], the authors propose a new method for class incremental learning that improves continual learning methods based on knowledge distillation with additional projections to intermediate subspaces with geodesic flow and proposed loss _Geodesic Distillation Loss_. The novel approach, GeoDL, is then combined with the exemplar-based method, i.e., LUCIR, iCARL, where the improvement is presented. Please note that we consider exemplar-free scenarios and propose a much different, experts-based method. We also cannot see the clear relation mentioned by the Reviewer. To be precise, we cannot find measuring task or expert similarities in this work and any KL-divergence mentions.
> > >
> > > [3] presents a PAC-Bayesian analysis of lifelong learning under two types of relaxations of the i.i.d. assumption on the tasks: to non-i.i.d tasks and task-related over time, e.g., classifier task of increasing difficulty. One of the main outcomes of this work is a presentation that accumulating knowledge over the course of learning multiple tasks can be beneficial for the future even if these tasks are not i.i.d. This can be a good theoretical grounding for exemplar-free class-incremental learning. However, this is more a theoretical work, with a simple toy experiment to present the results. Here, KLD is used to check two distributions' similarity for PAC-Bayesian analysis: P and Q, which are probability distributions over the hypothesis set we learn to solve the tasks. The relation of the task is given (assumed) by the two scenarios. However, we utilize KLD for a different purpose: to alleviate forgetting in the ensemble by selecting the right expert to be trained with the current task data.
> > >
> > > The first major difference between [4] and SEED is that [4] considers task-free CL, which is different from class incremental learning considered in our work. Therefore, [4] assumes the data comes as a stream and that the learner may have access to old classes as they have the same probability of appearing in the stream as new classes. On the other hand, we, and other CIL methods, do not require access to past data. That leads to major design differences between SEED and [4]. Our expert consists of a feature extractor that outputs feature vectors and a set of Multivariate Gaussian distributions representing classes. In [4], the expert consists of a density approximator and a classifier that outputs logits. When new data arrives [4] adds a new expert or performs a simple fine-tuning of all experts, which in CIL would result in a high forgetting. On the contrary, we fine-tune only a single expert in each task and utilize knowledge distillation to alleviate forgetting. Finally, [4] has no upper bound for the total number of experts, where we limit their number, e.g., to 5 in our experimental section.
> > >
> > > Even if we combine parts from the above methods, they will not result in our approach. Here, we enlist unique parts of SEED:
> > >
> > >  1. We train only a single, selected expert on each task; [4] trains all of them. To the best of our knowledge, our method of selecting an expert to be trained is novel.
> > >  2. We limit the number of total experts, e.g., to 5 in our experiments.
> > >  3. We represent knowledge using multivariate Gaussian distributions.
> > >  4. We utilize an ensemble of Bayes classifiers for the inference.
> > >  5. We do not use KL-divergence to measure differences between tasks as in [1] because we have only access to data of one task at a time, the current one. We utilize KL-divergence to measure the overlap between representations learned by experts, and based on this, we select which expert to fine-tune on the new task. We show that this selection decreases forgetting of the expert.
> > >
> > > Those are the main points that can distinguish our work from any combination of [1],[2],[3], and [4]. We hope we have addressed the Reviewer's concern about novelty and its relation to given references. We kindly ask to reconsider our score. Otherwise, we are still available and ready to discuss any questions and issues about our method.

---

> > > > ### Comment · Reviewer_rq4G · 2023-11-21
> > > >
> > > > Thank you for your explanation. I have updated my rating accordingly.

---

> > > > > ### Author Response · Authors · 2023-11-22
> > > > >
> > > > > One more time, we would like to express gratitude to the Reviewer for their time spent on the reviewing process, insightful comments, improving the quality of our work and our score. Thank you.

---

### Official Review · Reviewer_hfuR · 2023-10-30

**Soundness:** 3 good
**Presentation:** 3 good
**Contribution:** 2 fair
**Rating:** 6
**Confidence:** 3

**Summary:**

This work studies the class incremental learning (CIL) problem from the mixture-of-experts (MOE) angle where only a subset of the model (expert) is activated for each task. To this end, the authors propose SEED that selects one expert per task during continual training and aggregates the experts during evaluation. Thus, SEED promotes diversity representations while requiring no task identifiers during inference. Experiments on various CIL benchmarks show promising results of SEED.

**Strengths:**

- This work studies the MoE for CIL, which has been gaining interests as a promising approach to CIL.
- The proposed method is simple yet demonstrated encouraging results.
- The authors conduct various ablation studies to explore different aspects of SEED.

**Weaknesses:**

## Major concern - experiment
- In table 1 and 2, although it is clear that SEED performs better than the baselines, different methods have different memory footprint (model parameters and other components) or training complexities. Thus, it is difficult to judge if the gains come from the proposed method or from the additional complexities.

## Major concern - conceptual drawback of SEED
- SEED implies that the number of experts should be smaller than the number of tasks and there is only one expert activated per task. Thus, it is inevitably that some experts will be reused in the future, leading to catastrophic forgetting especially when the tasks are conflicting. The use of LwF regularization might not be helpful if the expert stay inactive for a long period.
Together with with the ambiguity in the experimental settings and results, the contribution of this work seems marginal.

## Minor concern
- Additional baselines: A recent related method [A] that seems to outperforms SEED should be discussed.
- There are two different references for CoSCL in the Introduction.

[A] Ardywibowo, Randy, et al. "VariGrow: Variational Architecture Growing for Task-Agnostic Continual Learning based on Bayesian Novelty." International Conference on Machine Learning. PMLR, 2022.


## After Rebuttal
After the rebuttal, the authors addressed most of my concerns and I adjust the rating accordingly. In the final version, the authors are strongly encouraged to clearly state the assumptions and scenarios that the proposed method will work well on.

**Questions:**

- Performance comparison under a fair setting.
- How could SEED avoid forgetting when the number of tasks is significantly larger than the number of experts.

---

> ### Author Response · Authors · 2023-11-15
>
> We thank the Reviewer for the constructive feedback, insightful comments and providing a reference. Below we respond to the weaknesses mentioned:
>
> **Do the gains come from the proposed method or from the additional complexities?**
>
> We agree that the question whether the gains come from the proposed method or the additional complexities is important. Therefore, we run the experiments with results presented in Tab. 3 and Tab. 4 of our submission. More specifically, in Tab. 4, we show that our method of ensembling experts in CIL outperforms existing SoTA ensembling methods; all require the same amount of parameters. In Tab. 3, our method achieves better results than competitors while using fewer parameters. We have also performed another experiment presented in Tab. 5 (and below) where we evaluate methods with regard to the number of theirs parameters during inference. We have added it to the revised version. The above observations lead us to the conclusion that the gain of our method does not result only from the additional complexities.
>
> We affirm that various CL methods present different memory/computational footprints. Especially for ensemble methods, it is inherent that few experts have to be stored in memory. Evaluating the complexity of some methods from Tab. 1 & 2 is not trivial, e.g., SSRE uses a complex few-phase training with expansion and pruning phases. However, pruning can also be applied to every other method from these tables. We try to be clear about SEED, and therefore we present and discuss: (1) the parameter efficiency in Tab. 3 compared to other MoE and ensemble architectures, (2) memory footprint in Appendix A.3, (3) we have also added a tradeoff between number of models' parameters vs accuracy in Appendix A.4 (revision version) to better address this concern.
>
> **Performance comparison under a limited number of parameters.**
>
> To address this concern, we have added Tab. 5 to the appendix (and below). SEED achieves better accuracy than the best competitor (FeTriL) from Tables 1&2 while using less parameters. Here we consider number of parameters required by the method to perform the inference. For SEED these paramters come from feature extractors of experts (here they share first 17 layers) and Gaussian distributions.
>
> | Method | Network    | Network weights |Total parameters| \|T\|=10 | \|T\|=20 | \|T\|=50 |
> | -------- | -------- |-------- |-------- | -------- | -------- | -------- |
> | EWC     |ResNet32   |  473K   |  473K  | 24.5     | 21.2 | 15.9
> | LWF     |ResNet32   |  473K   |  473K  | 45.9     | 27.4 | 20.1
> | FeTrIL  |ResNet32   |  467K   |  473K  | 46.3     | 38.7 | 27.0
> | SEED    |ResNet20   |  339K   | 460K   | 54.7     | 48.6 | 33.1
>
> **Avoiding catastrophic forgetting in SEED**
>
> One of the main goals of our method is to limit catastrophic forgetting. For that reason, we proposed our expert selection strategy to alleviate forgetting. We demonstrate the improvements in Tab. 4 and Fig. 6. Please note that we train only one expert in each task and that knowledge distillation regularization is applied only to it. If an expert is inactive (not trained on tasks), the parameters of its feature extractor do not change. Thus, it's not prone to catastrophic forgetting.
>
> **How could SEED avoid forgetting when the number of tasks is significantly larger than the number of experts?**
>
> In that case SEED avoids forgetting in two ways. First, in each task SEED trains only one expert, other experts (K-1) remain unchanged so that they avoid forgetting. Please note, that layers shared by experts are frozen after the first task. Second, we utilize feature distillation for trained expert to alleviate its forgetting - we made it more clear in Section 3 in the revised version.
>
>
> **Comparing VariGrow to SEED**
>
> The method cited by the Reviewer performs experiments with exemplars and auxilary datasets (4.1 Implementation details: ```We also use these exemplars along with OoD datasets```), while we propose an exemplar-free method for more challenging scenarios and we do not use exemplars or auxilary data in our experiments.
>
> **Two different references for CoSCL in the Introduction.**
>
> We have fixed this in the revised version. Thank you.
>
> ---
>
> We trust that our explanation has addressed the Reviewer's concerns. Should there be any additional queries, we are more than willing to provide further details. If no further clarification is needed, we kindly ask the Reviewer to reconsider the final score.

---

> ### Comment · Reviewer_hfuR · 2023-11-17
> **Conceptual Drawback of SEED**
>
> I appreciate the authors efforts in addressing my comments.
>
> However, my concerns regarding the drawback of SEED remains. SEED alleviates forgetting by freezing the feature extractor and activating only 1 expert per task. However, this strategy strongly assumes that the first task can provide a good feature representation for all future tasks and there are no tasks that require a completely different representation, which are quite restrictive. For example, consider the complex transfer settings in the CTrL benchmark [A] where the data stream can have unrelated tasks ($\mathcal{S}^{\text{plastic}}$), or the first task only have limited training data ($\mathcal{S}^+$), then SEED may not perform well on such scenarios.
>
> It may not be feasible to conduct comprehensive experiments on the CTrL benchmark during this rebuttal period, but the authors should clearly state the assumption of SEED, indicating which settings it might work well on, and acknowledging the conceptual drawbacks.
>
> [A] Veniat, Tom, Ludovic Denoyer, and Marc'Aurelio Ranzato. "Efficient continual learning with modular networks and task-driven priors." ICLR 2021.

---

> > ### Author Response · Authors · 2023-11-17
> >
> > We express our gratitude to the Reviewer for the quick and insightful response. Please find our answers below.
> >
> > **CTrL benchmark evaluation**
> >
> > We thank the Reviewer for pointing out this benchmark. Although, as rightfully mentioned by the Reviewer, time does not permit us to run the evaluation on this framework within the rebuttal period, we acknowledge the need to state the assumptions of SEED clearly and reference CTrL work as a relevant part of the literature. We have done so, revising the manuscript as described below.
> >
> > In Section 2 we have referred to CTrL after introducing FeTrIL work:
> > > However, freezing the backbone can limit the plasticity and not be sufficient for more complex settings, e.g., when tasks are unrelated, like in CTrL (Veniat et al.,2020).
> >
> > In Section 5 we have added the new limitation:
> > > Firstly, SEED may be not feasible for scenarios where tasks are completely unrelated and the number of parameters is limited, as in that case sharing initial parameters between experts may lead to a poor performance.
> >
> > **SEED in complex transfer learning scenarios**
> >
> > In these scenarios, we recommend that SEED shares as few initial frozen layers $f$ between experts as possible. However, we agree that this increases the memory footprint of the method and may be a limitation, which we have addressed above.
> >
> > Please note that we tried to simulate such scenarios in Tab.1 using DomainNet - where tasks are of different domains, and utilizing CIFAR100 split for 50 tasks, where each task consists just of two classes (small first task scenario). SEED achieves better results than competitors in these cases, even when the number of parameters is considered (Tab. 5).
> >
> > **Not freezing the whole feature extractor**
> >
> > In response to the remark about the extractor, we would like to clarify that SEED does not freeze the whole feature extractor, only layers $f$. As stated in Section 3 and caption of Fig. 2:
> > >SEED comprises $K$ deep network experts $g_k \circ f$, sharing the initial layers $f$ for higher computational performance, which are frozen after the first task.
> > >
> > In our experiments, we never freeze all layers of the feature extractor. To improve the clarity of our setting, we have changed the above sentence in the revised version to:
> >
> > >SEED comprises $K$ deep network experts $g_k \circ f$, sharing the initial layers $f$ for higher computational performance. $f$ are frozen after the first task.
> >
> > Therefore, when an expert $i$ is trained on a given task, we train layers $g_i$.
> >
> > **Regarding alleviating forgetting**
> >
> > Please note that freezing part of SEEDs parameters is not the only way we alleviate forgetting. We also limit it by selecting an expert, for which it will be the least, to be trained on a given task. For this, we utilize a novel expert selection strategy, whose results are presented in Fig. 6. Moreover, in the trained expert, we utilize feature distillation to alleviate its _personal_ forgetting.
> >
> > ---
> >
> > Once again, we thank the Reviewer for the remarks and are happy to discuss any other questions or comments.

---

> > ### Author Response · Authors · 2023-11-21
> >
> > Dear Reviewer hfuR,
> >
> > The discussion period ends in 24 hours.
> >
> > Once again, we thank you for your valuable insights into our work.
> >
> >
> > We have made every effort to address your concerns in our revised work by conducting more experiments and additional clarifications. Thanks to the Reviewer, we have improved our work. We kindly request that you consider these in your final assessment and consider raising your score in your final response.
> >
> > Please do not hesitate to let us know if there are additional points we can address.

---

> ### Author Response · Authors · 2023-11-22
>
> Although the rebuttal period is almost over, we are still eager to answer the Reviewer's questions and continue the discussion. Insights provided by the Reviewer were helpful and allowed to increase the quality of our work. If we have addressed uncertainties regarding our work, we would appreciate adjusting the final score.

---

> > ### Comment · Reviewer_hfuR · 2023-11-22
> > **Acknowledgement**
> >
> > Dear authors,
> >
> > I have gone through the discussion thus far. Although I have no further questions, I would like to discuss with AC and other Reviewers before finalizing my rating.
> >
> > Thank you

---

> > > ### Author Response · Authors · 2023-11-22
> > >
> > > We thank the Reviewer for the quick response. If any new questions regarding our method appear, we are still available to discuss them in detail.

---

### Official Review · Reviewer_3XPS · 2023-10-31

**Soundness:** 3 good
**Presentation:** 4 excellent
**Contribution:** 3 good
**Rating:** 8
**Confidence:** 4

**Summary:**

The authors propose a new class-incremental learning algorithm which relies on a set of expert models with a shared backbone. In each training step, only one expert is trained on the new task. This expert is chosen based on how well it separates the new classes. At inference time, all experts make predictions but their contribution to the final prediction is conditioned on the input.
Experiments are conducted on three different datasets in two variants and for task-agnostic and task-aware cases.
The authors provide several ablation studies on core components of the method.

**Strengths:**

The authors make an interesting observation by showing that some methods work significantly better if more data is available initially.
They propose a method that improves over the baselines, in particular in the case where few data is available initially. The method and description is good and reminds me of S-Prompts which has a similar setup but works with pretrained transformer models.
As far as I can tell, all important aspects are covered by ablation studies, leaving only room for few questions.

**Weaknesses:**

A not fully covered discussion is the number of parameters vs accuracy tradeoff. The proposed methods requires significantly more parameters than some of the baselines which might be a limitation if the models are extremely large. Furthermore, training from scratch is a rather uncommon scenario. It is unclear how this method would work if a pretrained model is used.

**Questions:**

Figure 7 (right): what is the difference between SEED with 1 expert and finetuning? What is your explanation that your method is doing very well in this particular setup?

---

> ### Author Response · Authors · 2023-11-15
>
> We express our gratitude to the reviewer for the constructive feedback and insightful remarks. We shall commence by addressing the specific weaknesses pointed out:
>
> **The tradeoff between number of parameters and accuracy**
>
> We have run additional experiments and presented them in Appendix A.4, where we use ResNet32 and ResNet18 architectures and provide accuracy depending on the number of experts and shared layers. The tradeoff between an overall number of model parameters and accuracy can be observed. We can use that analysis to limit the parameters and still reach a satisfactory accuracy. However, we agree that number of parameters can be a limiting factor our method if the models are extremely large, as few experts have to be stored in memory.
>
> **The impact of a pretrained model on SEED**
>
> We have performed experiments to examine the impact of using a pretrained model vs training from scratch. The results for DomainNet datasets are presented below (we also added them in A.5):
>
> | DomainNet \|T\| | Avg. acc. scratch | Avg. acc. pre | Avg. forgetting scratch | Avg. forgetting pre |
> | :--------: | :--------: | :--------: | :--------: | :--------: |
> | 12     | 45.0     | 53.1     | 12.1 | 12.8
> | 24     | 49.0     | 54.2     | 11.2 | 12.1
> | 36     | 39.2     | 53.6     | 13.7 | 15.6
>
> We use DomainNet here as we consider it the most interesting scenario to use ImageNet pretrained network. SEED can benefit from starting each expert from the pretrained network and reach better accuracies with a slightly higher forgetting. However, here, we use the same setting as from the scratch (hyperparameters), which can be suboptimal for this scenario.
>
> **Differences between finetuning and one expert setting**
>
> There are two significant differences between using SEED with one expert and finetuning. First, we perform Bayes classification during the inference. Second, we use a feature distillation to increase the stability of the expert. The method is doing well in this scenario as in our ensemble, few experts are not finetuned and increase stability of the method, while the one being finetuned allows for good plasticity.
>
> ---
>
> If we have adequately addressed the Reviewer's concerns, we kindly ask for your support. If you have any further concerns or additional points to raise, we are eager to address them. Your insights are valuable in enhancing the quality and impact of our research.

---

> > ### Comment · Reviewer_3XPS · 2023-11-17
> >
> > Thank you for addressing my comments. I have no further questions.

---

> > > ### Author Response · Authors · 2023-11-21
> > >
> > > Thank you for considering our paper worth publication - we appreciate all the constructive feedback that allowed us to improve our submission.

---

### Official Review · Reviewer_Hjf8 · 2023-11-02

**Soundness:** 3 good
**Presentation:** 3 good
**Contribution:** 3 good
**Rating:** 8
**Confidence:** 3

**Summary:**

The paper proposes an (exemplar-free) ensemble method for Class Incremental Continual Learning (CIL). A fixed number of experts are trained on a stream of tasks. At each task, only one expert is finetuned. The result is a diverse expert ensemble which is reported to perform very well on several benchmarks.

**Strengths:**

I believe this paper advances the field of continual learning.

Specifically,

- Tackles a popular continual learning scenario.
- Good empirical results. Show significant improvements on several datasets.
- Well written and clear.
- Sound experiments with several ablation studies.

**Weaknesses:**

No major weaknesses. Perhaps just the fact that several models (experts) have to be trained and stored (but this is inherent in ensemble methods).


### Minor remarks (I don't expect any response on these remarks in the rebuttal)
- In Figure 3, it's hard to see that task 3 overlap least with the second expert. Perhaps make this a bit clearer.
- In Page 4, perhaps the inference and training algorithms would be easier to read in an "Algorithm" structure rather than simple paragraphs.
- In Page 5, just before Section 4.1., the task incremental scenario is mentioned but is not really defined properly in my opinion.

**Questions:**

1. In Page 8, it's written that *"SEED uses a regularization method known from LwF"*. Are you actually using LwF for each expert? I don't recall reading it in other places in the paper. Please clarify.
1. How are the hyperparameters of EWC and LwF set in the experiments ? For instance, Figure 7 shows $\lambda\in [100,10000]$ without any justification.
1. In Page 1, the authors mention *"The trend is evident... results steadily improve over time"*. What time?
1. In Table 1, how many repetitions are performed per experiment?
1. In Table 1, why aren't there results for ImageNet-Subset for two algorithms?
1. In Figure 5, the presented metric is "relative accuracy". Relative to what?

---

> ### Author Response · Authors · 2023-11-15
>
> We thank the Reviewer for the positive feedback and insightful comments. We begin by responding to the weakness pointed out in the review:
>
> **The need to train and store a number of experts**
>
> As the Reviewer stated, this is inherent in ensemble methods. However, we discuss the number of parameters in Tab.3 and Appendix A.3. We have additionally run experiments to examine the trade-off between the number of parameters and accuracy of SEED in Appendix 4. We show that the total number of parameters can be lowered, and SEED can still achieve satisfactory results.
>
> **Improvements in the paper**
> >_In Figure 3, it's hard to see that task 3 overlap least with the second expert. Perhaps make this a bit clearer._
>
> We appreciate that suggestion. We updated the figure in the new revision.
>
>
> >_In Page 4, perhaps the inference and training algorithms would be easier to read in an "Algorithm" structure rather than simple paragraphs._
>
> >_In Page 5, just before Section 4.1., the task incremental scenario is mentioned but is not really defined properly in my opinion.
>
> We have addressed both comments in the revised version. Thank you.
>
> **Questions**
>
> >_In Page 8, it's written that "SEED uses a regularization method known from LwF". Are you actually using LwF for each expert? I don't recall reading it in other places in the paper. Please clarify._
>
> SEED uses feature distillation for each expert, similar to LwF (logit distillation) but performed in the latent space. It's mentioned in Sec. 3, Training (page 4/5). We clarified this part of our method description in the new revision.
>
> >_How are the hyperparameters of EWC and LwF set in the experiments ? For instance, Figure 7 shows without any justification._
>
> We set default parameters of the methods as reported in the original articles. For avoidance of doubt, we also stated it in Appendix A.1: ```For baseline methods, we set default hyperparameters```. For Fig. 7 hyperparameters were manually chosen to exhibit the tradeoff between intransigence and forgetting, and we list all these parameters in Appendix A.1 as well.
>
> >_In Page 1, the authors mention "The trend is evident... results steadily improve over time". What time?_
>
> In this case, we meant the dates of publication of the corresponding articles. The methods presented in Fig. 1 are positioned according to the order of their publication dates. We have labeled the publication year axis x in Fig. 1 to better visualize that in the revised version.
>
> >_In Table 1, how many repetitions are performed per experiment?_
>
> We perform three repetitions per experiment, as stated in Appendix A.1: ```All experiments are the average over three runs```.
>
> >_In Table 1, why aren't there results for ImageNet-Subset for two algorithms?_
>
> For these two methods, we performed experiments using their implementations in PyCIL benchmark and default hyperparameters. However, on ImageNet-Subset IL2A performance was really low, while PASS experiments did not converge, probably getting stuck, and we could not finish them.
>
> >_In Figure 5, the presented metric is "relative accuracy". Relative to what?_
>
> We present the metric relative to the mean achieved by all experts. We have clarified that in the caption of Fig. 5 in the updated version: ```It is calculated by subtracting the accuracy of each expert from the averaged accuracy of all experts.```
>
> ---
>
> If the Reviewer's concerns have been sufficiently addressed in our responses, we humbly seek their support for the paper.

---

> > ### Comment · Reviewer_Hjf8 · 2023-11-21
> >
> > Thank you for addressing my comments.
> >
> > Regarding my minor remarks - in the revised Figure 3, I still cannot see immediately that the purple task overlaps least with the second expert. Regarding the algorithm in Page 4, I suggested using an [algorithm environment](https://www.overleaf.com/learn/latex/Algorithms), not just writing "Algorithm" in the paragraph's title.
> >
> > Anyway, I have no further questions and I keep my score (8).

---

> > > ### Author Response · Authors · 2023-11-21
> > >
> > > One more time, we express our gratitude towards the Reviewer for improving the quality of our work and considering our work to be worth publication. We have improved Fig. 3 in the newest revision.

---

### Author Response · Authors · 2023-11-15

We would like to thank all the Reviewers and the Chairs for their hard work. We have carefully read the reviews and addressed all the issues raised, which undoubtedly improved the quality of our submission.

We uploaded the revised version of our work and are ready to participate in further discussion. Based on comments, we have made the following changes to the revised version:
- To better analyze the trade-off between accuracy and parameters in SEED and examine its performance under a limited number of parameters, we have performed new experiments. We provide and discuss them in Appendix A.4.
- To investigate the impact of using pretrained models in SEED we have conducted additional experiments and present them in Appendix A.5.
- To improve readability, we have updated Fig. 1 and Fig. 3.
- For method clarification, we provide the knowledge distillation loss equation in Section 3.
- In the caption of Fig. 5 we describe the relative accuracy metric used in the plot in more detail.

We thank the Reviewers once again and look forward to discussing any other aspects of the paper that require further clarification.
\--- authors

---

### Meta-Review · Area_Chair_2uTa · 2023-12-09

**Metareview:**

In this paper, the authors propose a novel continual learning method that maintains a set of expert models. Overall, this paper is well-organized and easy to read. The authors also provide extensive experimental results with impressive performance. After the rebuttal session, all reviewers are positive towards this paper. Therefore, I recommend acceptance.

**Justification For Why Not Higher Score:**

This paper provides interesting results. However, some weaknesses are not fully solved in the current manuscript. For example, some recent popular continual learning methods, such as [FOSTER](https://arxiv.org/abs/2204.04662) and [DER](https://openaccess.thecvf.com/content/CVPR2021/papers/Yan_DER_Dynamically_Expandable_Representation_for_Class_Incremental_Learning_CVPR_2021_paper.pdf) are not directly compared in the main tables. Therefore, the final recommendation is a poster.

**Justification For Why Not Lower Score:**

All the reviewers tend to accept this paper. Considering the quality of this paper, it is definitely a clear acceptance.

---

### Decision · Program_Chairs · 2024-01-16

Accept (poster)